# Shear Wave Elastography-Correlated Dose Modifying: Can We Reduce Corticosteroid Doses in Idiopathic Granulomatous Mastitis Treatment? Preliminary Results

**DOI:** 10.3390/jcm12062265

**Published:** 2023-03-15

**Authors:** Bunyamin Ece, Sonay Aydin, Mecit Kantarci

**Affiliations:** 1Department of Radiology, Kastamonu University, Kastamonu 37150, Turkey; bunyaminece@hotmail.com; 2Department of Radiology, Erzincan University, Erzincan 24100, Turkey; akkanrad@hotmail.com; 3Department of Radiology, Atatürk University, Erzurum 25240, Turkey

**Keywords:** idiopathic granulomatous mastitis, shear wave elastography, corticosteroid, tissue stiffness, elasticity value

## Abstract

Idiopathic granulomatous mastitis (IGM) is a chronic inflammatory breast disease treated with local and systemic corticosteroids. This study aims to evaluate the efficacy of reducing corticosteroids doses in IGM cases based on shear wave elastography (SWE) tissue stiffness measurements. This prospective study included IGM patients who received systemic or local corticosteroids between January 2020 and September 2022. A 20% or more reduction in tissue elasticity values (kPa) was considered a positive response to treatment in the study group, and the corticosteroids dose was reduced. The control group was dosed routinely. All patients were followed for 2 years to compare treatment efficacy, duration, total corticosteroids dose, recurrence, and side effects. There were 12 patients (9 local/3 systemic corticosteroids) in the study group and 24 patients (17 local/7 systemic corticosteroids) in the control group. Ten (83.4%) out of 12 patients in the study group were successfully treated by reducing corticosteroid doses with follow-up, and 2 (16.6%) out of 12 patients were reverted to the initial treatment protocol due to an increase in elasticity values during the follow-up. Nevertheless, successful treatment results were obtained in these two patients without reducing the corticosteroid dose. When compared to the control group, the median corticosteroid dose in the study group was significantly lower in patients using both local (*p* < 0.01) and systemic (*p* < 0.01) corticosteroids. A significant negative correlation was found between the rate of decrease in elasticity values and the median dose of corticosteroids (r = −0.649, *p* < 0.05) and the median treatment time (r = −0.751, *p* < 0.01). Side effects due to corticosteroids were found to be significantly lower in the study group (*p* < 0.05). According to our first and preliminary results, the SWE-correlated dose-modifying technique may reduce corticosteroid doses and side effects without significantly compromising treatment efficacy.

## 1. Introduction

Idiopathic granulomatous mastitis (IGM) (nonpuerperal mastitis or granulomatous lobular mastitis) is an uncommon benign chronic inflammatory breast illness first identified by Kessler and Wolloch in 1972. The high prevalence of IGM in certain ethnic regions and groups suggests that environmental and genetic factors may play a role in the disease’s etiology; the majority of IGM cases have been reported from Asian countries, particularly Turkey [1]. IGM is distinguished by sterile lobulocentric granulomatous inflammation. The natural illness course is usually repeated or extended. There are few data on the prevalence and incidence of IGM based on ethnicity; nevertheless, anecdotal evidence and the few instances documented in the literature imply that the condition is uncommon [2]. Both clinically and radiologically, it can mimic breast cancer. Imaging approaches have not yet achieved sufficient diagnostic accuracy to differentiate between the two. As a result, an accurate diagnosis requires a pathological examination. An inflammatory reaction disturbing mammary lobules and several noncaseating granulomas in the samples are used to make the final diagnosis [3].

Unnecessary biopsies or surgical excision can result in a persistent fistula and breast abnormalities. Prior to the 1980s, most IGM patients were treated solely with broad surgical excision. Conservative therapy, such as oral corticosteroids or imaging surveillance, is now recommended as a first-line therapeutic option before considering surgery [2,3]. Steroids are currently used locally to treat a variety of inflammatory and noninflammatory illnesses across the body, including carpal tunnel syndrome, cervical or lumbar spine discomfort, and diabetic macular edema. A wide surgical excision is also a possibility for the treatment of IGM. It has been established that the complete remission rates following surgical, oral steroid, and combined use of surgery and oral steroid treatment are 90.6%, 71.8%, and 94.5%, respectively, and the recurrence rates are 6.8%, 20.9%, and 4%, respectively, in a meta-analysis of 15 trials. The authors reported that surgical treatment was associated with a high percentage of full remission and a low rate of recurrence with or without steroid medication [4,5]. 

There are other studies indicating that steroid therapy prior to surgery decreases the incidence of recurrence. A surgical procedure performed on the inflammatory tissue may result in delayed wound healing and abscess and fistula formation, resulting in poor, undesirable cosmetic effects. Preoperative steroid treatment can reduce the size of large lesions and improve cosmetic outcomes. Systemic corticosteroid treatment has been established in studies to be a suitable and effective treatment option for providing full recovery and preventing long-term illness relapse. Long-term high-dose corticosteroid use, on the other hand, has significant adverse effects [4,5]. Considering this, local corticosteroid treatment was recently demonstrated to be equally effective as systemic corticosteroid therapy [5].

Elastography is a new supplementary imaging technology based on ultrasound (US) that improves the diagnostic performance of B-mode US by assessing tissue stiffness. Shear wave elastography (SWE) and strain elastography are the two types of elastography employed in breast lesion evaluation. Strain elastography is operator-dependent. SWE, on the other hand, uses focused radiation forces without manual compression and is not operator-dependent [6,7]. SWE has been shown to be useful for differentiating benign breast lesions from malignant breast lesions, and it has been suggested that SWE enhances the diagnostic performance of ultrasonography, potentially improving the specificity of conventional ultrasonography using the Breast Imaging Reporting and Data System criteria. Recently, SWE has not only been demonstrated to be beneficial for the diagnosis of breast cancer, but also to provide crucial information that can be utilized to forecast the prognosis or response to chemotherapy prior to surgery. The application of elastography, particularly SWE, in IGM is quite restricted; current research is focused primarily on the ability of elastography to distinguish malignancies from IGM [3,8].

In IGM cases, we use both local and systemic corticosteroid treatment in our daily practice. We have noted that the elasticity values (kPa) of the healing tissue reduce during the therapy and follow-up phase. We hypothesized that by employing elasticity values, we may alter corticosteroid doses, reducing adverse effects and unnecessary injections. The current study sought to assess the efficacy of SWE for the adjustment and monitoring of corticosteroid therapy in IGM cases.

## 2. Materials and Methods

The local ethics committee gave its approval for the current prospective cohort study (Ebyu-kaek-2020/12-6). Informed consent was acquired from all of the participants. The study was conducted between January 2020 and September 2022. 

### 2.1. Participants

During the study period mentioned above, 48 patients with biopsy-confirmed diagnosis of granulomatous mastitis who could be treated with steroids and who agreed to participate in the study were included. The exclusion criteria of this study were any contraindication for systemic and/or local corticosteroid use, malignancy, immunosuppression, being in the active lactation period, cessation of treatment due to the patient’s will or for medical reasons, and lack of follow-up data. According to the exclusion criteria of the study, 12 patients were excluded. 

A total of 36 female patients who received a diagnosis of IGM from a tru-cut biopsy were included in the study. The mean age of the patients was 38.0 ± 3.4 years. 

Patient group: Twelve patients with a diagnosis of IGM who were willing to participate were included in the study. The mean age of the study group was 37.8 ± 2.4 years. 

Control group: Twenty-four patients with a diagnosis of IGM were included in the control group. The control group consisted of spontaneously selected IGM patients with similar age and initial mean elasticity values. The mean age of the control group was 38.1 ± 3.7 years. The study groups can be seen in detail in Figure 1.

All the included patients were followed up for 2 years and final outcomes were compared. 

Elastographic imaging technique: High-frequency linear-array transducers (L12-3, 3–12 MHz) in the longitudinal and transverse planes were used to perform examinations (Affiniti 70 g Philips Healthcare, Best, The Netherlands) (Figure 2). The examinations were carried out in a supine position. Elasticity values were determined automatically by the device’s SWE feature. During ultrasonographic imaging, care was taken to avoid applying pressure to the probe and to keep the practitioner’s hand fixed. Elasticity values were obtained with a maximum shear wave velocity cut-off of 5 m/s (80 kPa) to improve specificity. We expressed the elasticity data in kilopascals (kPa), which is a derived SI unit of shear modulus. Shear modulus can be defined as the ratio of shear stress and shear strain [9,10]. The elasticity value can be expressed in meters per second (m/s or converted to kilopascals (kPa). The elasticity values of the lesions were measured three times with three different ROIs of 1 cm^2^ from the different areas by the same observer, and the average of these measurements was recorded as the final data. We used the radial scanning method both to find the lesion easily in later examinations and for a different practitioner to understand the described localization clearly and accurately. We recorded the lesions’ quadrant, precise location (at breast “o’clock”), size, and nipple-to-lesion distance. Due to the tissue changes secondary to the treatment and the time elapsed, it was not possible to match the previous measurement localization exactly. However, control scans were performed on the mastitis areas by the same person with the same number of measurements.

### 2.2. Treatment

Stable treatment method: Two types of treatment used were used in this study: local and systemic [5]. There were no patients who received both treatments. Following the drainage of any loculated fluid, a 21G needle was used to inject 5 mL of saline and 1 mL of a 40 mg methylprednisolone acetate solution into the perilesional regions under sonographic guidance, and 0.8 mg/kg/d methylprednisolone acetate was given orally for the systemic treatment. All of the patients were monitored on a monthly basis. The treatment was continued until the patient was completely recovered (complete recovery: no recurrence during 2-year period of follow-up). 

Assessment of treatment response and study protocol (dose-modifying technique): Before treatment, the patients’ mean elasticity values were recorded. In monthly follow-ups, a reduction in elasticity values of 20% or more was determined to be a positive response to treatment, and the corticosteroid dose was decreased accordingly. (The 20% reduction criterion is based on our clinical experience and observations. Since this is the first study in the literature to examine the dose-modifying method via SWE, we cannot find a similar and/or exemplary study in the literature to define a more objective reduction percentage).

For local treatment, the amount of injected solution was decreased to 4.5 mL-30 mg methylprednisolone instead of 6 mL-40 mg methylprednisolone. For systemic treatment, the daily dose was reduced to 0.6 mg/kg/d instead of 0.8 mg/kg/d methylprednisolone acetate (Figure 3). 

After first reducing the corticosteroid dose, patients were monitored on a monthly basis; if the treatment response persisted, the dose was reduced once more. The amount of injected solution for locally treated patients decreased to 3 mL-20 mg methylprednisolone instead of 4.5 mL-30 mg. For systemic treatment, the daily dose was reduced to 0.4 mg/kg/d instead of 0.6 mg/kg/d methylprednisolone acetate (Figure 3). 

If progression or cessation of treatment response (increased or similar control elasticity values) occurred in any phase of follow-up, the initial dosing regimen was reinstated [4] (Figure 3). 

Side effects: The patients were evaluated for the presence of steroid-related side effects during imaging controls with the participation of the relevant clinician, and steroid use-related side effects (nausea, joint pain, weight gain, and edema) during treatment have also been noted.

Statistical analysis: Using the SPSS package for social sciences (version 20) for Windows, the data were analyzed (IBM SPSS Inc., Chicago, IL, USA). To determine whether the data had a normal distribution, the Kolmogorov–Smirnov test was applied. Variables with a normal distribution are reported as the mean and standard deviation, whereas non-normal variables are reported as medians. For reporting categorical variables, numbers and percentages are used. Student’s T test was used to compare the study and control groups’ mean ages and elasticity values. The median corticosteroid dose and median treatment duration were compared using the Mann–Whitney U test. Using Spearman’s correlation analysis, potential correlations between age and elasticity values and the other parameters were examined. To compare percentages across groups, Fisher’s exact test was utilized. A value of *p* < 0.05 was regarded as statistically significant.

## 3. Results

Thirty-six female participants were included in the study (12 in the study group and 24 in the control group). The mean age of the participants was 38.0 ± 3.4 years. All of the included patients were successfully treated, and no recurrence (the recurrence of the pathology during the 2-year follow-up period after a complete recovery) was detected during the follow-up period. According to the exclusion criteria of the study, 12 patients were excluded. 

The mean age of the study group was 37.8 ± 2.4 years and that of the control group was 38.1 ± 3.7 years. There was no statistically significant difference between the mean ages. We could not detect any correlation between the age and elasticity values, decrease in elasticity values, median corticosteroid dose, and median time of treatment. 

There were 12 patients in the study group; detailed results of the study group are presented in Table 1. Of the 12 patients, 2 patients (Table 1, patients 8 and 9) experienced increased elasticity values during follow-up and we returned to the initial treatment protocol (2/12, 16.6%). These patients were all receiving local treatment. We did not encounter any unresponsiveness to treatment in systemically treated patients. 

In the control group, 7 patients received systemic and 17 patients received local treatment. For the locally treated patients, the median corticosteroid dose was significantly lower in the study group (130 mg vs. 200 mg; *p* < 0.01), whereas the median treatment duration was significantly longer (24 weeks vs. 20 weeks; *p* < 0.05). For systemic treatment, both median corticosteroid dose (7308 mg vs. 12,105 mg; *p* < 0.01) and median treatment duration (18 weeks vs. 28 weeks; *p* = 0.01) were significantly lower in the study group. 

The mean initial elasticity value of the study group was 76.65 ± 12.82 kPa and that of the control group was 78.35 ± 9.46 kPa; there was no statistically significant difference between the elasticity values. Initial elasticity values correlate positively with median corticosteroid dose (r = 0.873, *p* < 0.01) and median treatment duration (r = 0.741, *p* < 0.01) for both groups and both treatment methods (systemic or local). Additionally, we found a significant negative correlation between the rate of the decrease in elasticity values before dose adjustment and median corticosteroid dose (r = −0.649, *p* < 0.05) and median time of treatment (r = −0.751, *p* < 0.01) in the study group. 

In the study group, we detected side effects in two patients (2/12, 16.6%), whereas in the control group, five patients (5/24, 20.8%) experienced side effects, and the difference was found to be statistically significant (*p* < 0.05). The rate of side effects was statistically significantly lower for locally treated (11.1% vs. 17.6%; *p* < 0.01) and systemically treated patients (33.3% vs. 28.5%; *p* = 0.01) in the study group (Table 2).

## 4. Discussion

We have designed a pilot study to determine the effectivity of corticosteroid dose adjustment via SWE in IGM. We have revealed that through the defined dose adjustment method, significant corticosteroid dose and treatment time reduction could be achieved. 

Diverse methods of treatment have been suggested for IGM, and there is still no consensus on the most appropriate treatment. Only clinical and sonographic follow-up has been recommended as a follow-up option [11]. On the other hand, complete surgical excision has been proposed as an effective treatment with complete remission and relatively low recurrence rates, with or without corticosteroid treatment. However, performing a large surgical excision or mastectomy for benign lesions that are partially self-limiting is generally regarded as a difficult decision for both the surgeon and the patient [11,12]. Additionally, medical treatment options have been investigated. In accordance with a recent guideline [11], antibiotics can be administered based on the results of bacterial testing and drug susceptibility testing. The use of antibiotics is strongly recommended, and the quality of the evidence supporting this recommendation is moderate.

Immunosuppression is proposed as the primary therapeutic mechanism. Non-corticosteroid immunosuppressive agents, such as methotrexate, can be used for patients who are resistant to corticosteroids or intolerant of their long-term side effects; however, this method is not strongly recommended and the quality of the evidence is low [11,13].

Both local and systemic corticosteroid use has been widespread in IGM. The administration of corticosteroids prior to surgery for large lesions improves the aesthetic outcome. Intralesional corticosteroid injection and topical corticosteroid can be used to treat IGM patients with primarily skin changes or who have experienced systemic corticosteroid side effects [11,14]. The principal drawback of corticosteroid therapy is its side effects, which manifest themselves primarily after long-term, high-dose use. In previous studies, a high dose of oral prednisolone was defined as >5 mg per day for more than one month. In addition, a daily dose of 60 mg has been associated with adverse effects [4,15]. Successful treatment outcomes have been reported at doses of 25 mg/d and 0.8 mg/kg/d [15,16]. Our corticosteroid doses are also within the limits previously reported. Numerous previous studies, as well as our own, focused primarily on decreasing corticosteroid doses to reduce side effects.

SWE is a relatively new, sonography-based technique used to evaluate tissue stiffness, and its use has been expanding. A region of interest (ROI) can be placed on a colored elastogram to measure the tissue elasticity value, which can be expressed in meters per second (m/s) or converted to kilopascals (kPa). IGM increases tissue rigidity, which can lead to confusion with malignant lesions. Elastographic studies of IGM have focused primarily on distinguishing these lesions from malignancy [3,8]. To the best of our knowledge, no previous study has examined the effects of treatment or manipulated treatment using SWE parameters.

In our study, the SWE-correlated active dose modification technique demonstrated promising benefits. Only two patients were forced to revert to the original treatment protocol due to increased elasticity values during follow-up. Both of these patients were receiving local treatment, but this cannot be interpreted as the presented dose-modifying method having a higher failure risk for local treatment in comparison with systematic treatment, since the number of systemically treated patients (three patients) was too small and may not be representative of the actual population. The SWE-correlated dose-modifying technique permitted a substantial reduction in total corticosteroid dose for both systemic and local treatment. Consequently, even though the total number of patients experiencing side effects was quite limited, the incidence of side effects decreased, leading us to hypothesize that the decrease in corticosteroid dose reflects the clinical course. On the other hand, the SWE-correlated dose-modifying technique slightly increased the duration of treatment for locally treated patients. This increase in the time of treatment means that patients will have more injections. As a result, despite a decrease in total dose and correlated side effects, patient compatibility could be negatively impacted by this dose modification. 

We have identified a correlation between elasticity values and disease progression. Higher elasticity values are associated with longer treatment durations and increased corticosteroid dosages. In addition, we have demonstrated a correlation between greater decreases in elasticity values and shorter treatment durations and lower corticosteroid doses. Aslan et al. found no correlation between the pre-treatment SWE findings and the severity of the IGM. This is the only similar study found in the literature, and its results were contradictory [17]. We believe that the main reason for the contradicting results is that patients in Aslan et al.’s study were treated with a variety of methods, not just with steroids. This can change the outcome of the treatment and produce results that contradict ours.

In two patients, the presented method was ineffective. Even though these patients were older than the mean age of the population, two patients are insufficient to draw a judgment concerning the unfavorable effect of age on the treatment approach. In addition, we were unable to establish any correlation between age and the remaining parameters. We cannot find any information in the literature regarding the effect of age on disease progression. Additional prospective research is required to clarify the possible relationships.

There are some limitations worth noting about the current study. The most significant limitation is the small sample size, especially for the systemic treatment group. The sample size is only sufficient for a pilot study. Additional studies with larger populations and more systemically treated subjects are needed, and would reveal different results. As a further consequence of the small sample size, all of the included patients were successfully treated, and no recurrent cases were observed. We were unaware of the effect of the presented method on cases of this nature. Due to the small number of patients for whom our method failed, we are unable to identify any potential flaws in our approach. Our clinical experiences were comprised of the treatment doses and rate of decrease in elasticity values to adjust the new dosing regimen. The results may vary depending on different dosages and protocols. Due to the fact that each SWE examination was conducted by the same researcher, we cannot provide data on inter-rater variability. The number of patients who develop side effects related to steroid use is relatively small. For this reason, the effect of the method we have presented on the side effect profile has been evaluated to be relatively limited.

## 5. Conclusions

In conclusion, the presented SWE-correlated dose-modifying technique may reduce corticosteroid doses and corticosteroid-related side effects without significantly compromising treatment efficacy. 

## Figures and Tables

**Figure 1 jcm-12-02265-f001:**
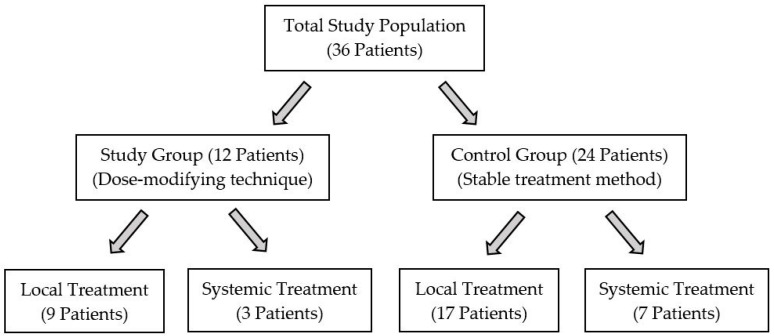
Grouping of participants and treatment methods.

**Figure 2 jcm-12-02265-f002:**
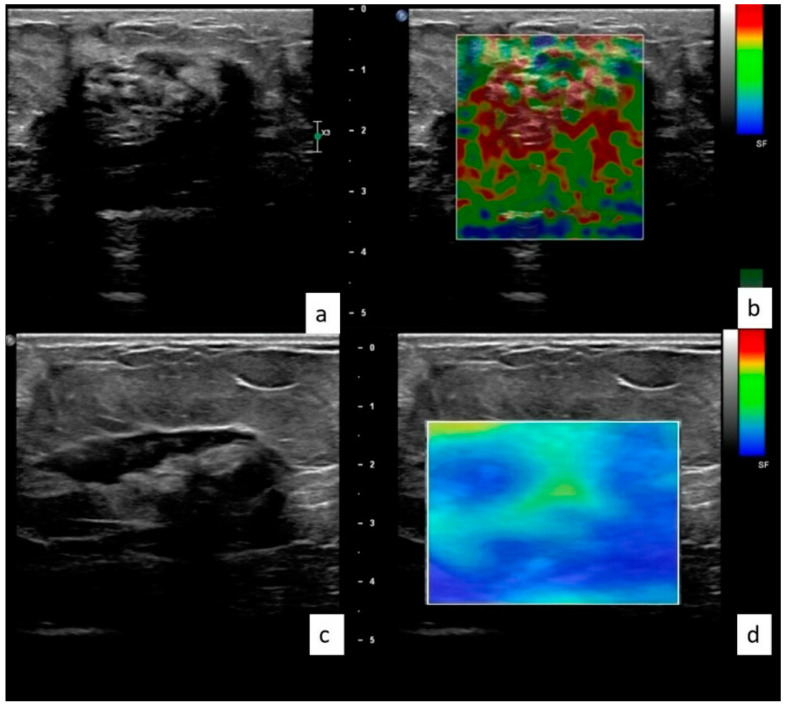
38 year old female with pathologically confirmed granulomatous mastitis. The lesion was located on the right breast, upper quadrant. The patient was treated locally. Elastographic examinations at the time of diagnosis (**a**,**b**) and 8 weeks after treatment (**c**,**d**) reveal that the lesion’s elasticity value (kPa) decreased by 57.9%, from 77.5 to 32.6 kPa.

**Figure 3 jcm-12-02265-f003:**
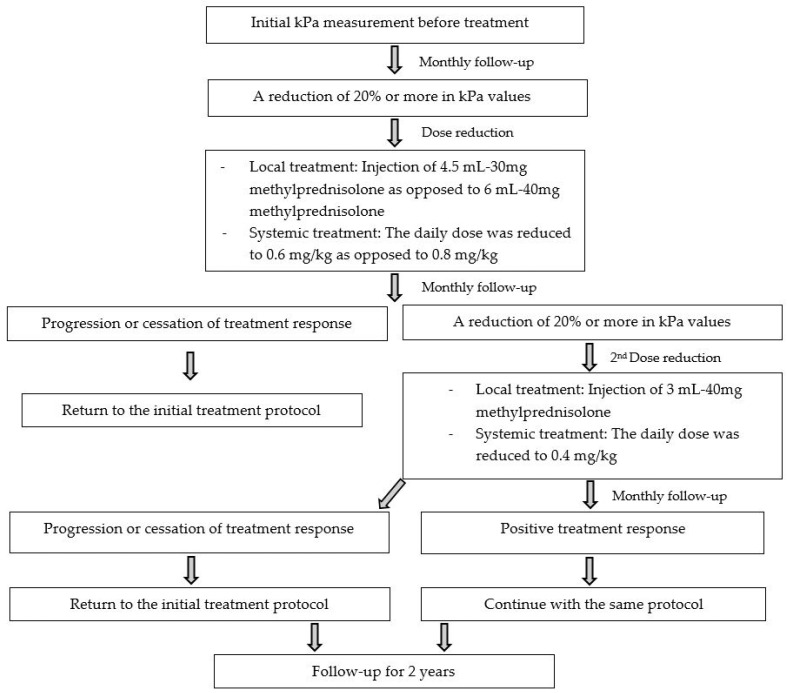
Assessment of treatment response and study protocol.

**Table 1 jcm-12-02265-t001:** Detailed information about study group.

Patient No:	Age	Systemic/Local Steroid	Initial kPa	After Treatment/Before Dose Adjustment kPa (% of Decrease)	Total Steroid Dose (mg)	Total Time of Treatment (Weeks)	Final Dosing Regime
1	39	Local	63.7	42.9 (32.6)	110	24	3 mL
2	46	Local	83.6	62.4 (25.3)	310	44	4.5 mL
3	32	Local	61.9	33.8 (45.3)	90	15	3 mL
4	37	Local	75.4	49.4 (34.4)	120	25	3 mL
5	38	Local	87.3	40.5 (53.6)	80	15	3 mL
6	38	Local	61.5	31.7 (48.4)	130	19	4.5 mL
7	38	Local	77.5	32.6 (57.9)	130	14	4.5 mL
8 *	39	Local	91.5	72.7 (20.5)	380	42	Return to the original regime
9 *	41	Local	88.7	63.6 (28.2)	280	36	Return to the original regime
10	35	Systemic	62.6	38.5 (38.4)	4032	12	0.4 mg/kg/d
11	33	Systemic	72.5	54.8 (24.4)	7308	18	0.6 mg/kg/d
12	38	Systemic	93.6	67.2 (28.2)	10,752	24	0.6 mg/kg/d

* Patients returned to original regime. kPa: kilopascals.

**Table 2 jcm-12-02265-t002:** Side effects according to groups.

Side Effects, Number (%)	Study Group (Patient No)		Control Group (Patient No)	
	Local (9)	Systemic (3)	Local (17)	Systemic (7)
Nausea	1 (11.1%)	0	2 (11.6%)	0
Joint pain	0	0	1 (5.8%)	0
Weight gain	0	1 (33.3%)	0	1 (14.2%)
Edema	0	0	0	1 (14.2%)

## Data Availability

The data that support the findings of this study are available from the corresponding author upon reasonable request.

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
