# Peer review of "Shear Wave Elastography-Correlated Dose Modifying: Can We Reduce Corticosteroid Doses in Idiopathic Granulomatous Mastitis Treatment? Preliminary Results"

_jcm, 2023, doi:10.3390/jcm12062265_

Round 1

Reviewer 1 Report

4%) patients in the study group, corticosteroids doses were reduced and successfully treated. In the study group, the median dose of corticosteroids was found to be significantly lower in patients both locally (p=0.006) and systemic 22 (p=0.007) corticosteroids.

Hard to understand these sentences, they need improved English language. First, the doses were reduced and successfully treated, I hope that all patients were successfully treated, but did you mean nevertheless successful? Please add this information. And I don’t understand the second sentence as well: In the study group, the median steroid dose was lower compared to control group?? Overall or just during the follow up? Was the dose reduced maybe?? Please clarify.

Side effects due to corticosteroids were found to be statistically significantly lower in the study group

Statistically significantly is too much, better e.g. found to be significantly lower.

Introduction:

Long-term high-dose corticosteroid use, on the other hand, has significant adverse 49 effects [3,4]. Considering that, recently, local corticosteroid treatment was demonstrated 50 to be equally effective as systemic corticosteroid therapy [4].

I’d like to ask the authors, if you could add an information (short only) about how the imaging was performed prior to elastography to monitor either treatment response or the mastitis itself. Thank you.

Line 62: Better reduce than diminish

the efficacy of SWE stiffness adjusted corticosteroid therapy 65 method in IGM cases.

Better: the efficacy of SWE for adjustment and monitoring of corticosteroid therapy in IGM cases.

M&Ms:

Could be more detailed overall, there are many information missing, e.g. in the patients section: Patients who received biopsy were included. There are no information about the population (Consecutive patients, only female, male too? How many were excluded due to criteria??

Language needs revision by a native speaker (especially the treatment section, e.g. were injected perilesional areas…that’s not English, injected into would be), same with administered.

The treatment was continued until the patient was completely recovered.

Or until the study was over? How long was it monitored? Or did all included patients recover in this time?

Please add the information about the used transducers (MHz?). And why did you only measure the kPa and not the Shear wave speed? How many measurements did you take in each case to assess kPa, did you calculate a mean value of various measurements or just a single measurement? Was there a quality indicator to measure? How did you know that the measurement was correct? Where did you measure, same area? Different quadrants of the breast? Just injection side? (Could falsify measurements due to scarring tissue). Please add these information.

Did the control group have IGM or no? And monthly control scans were performed always with the same number of measurements? Same area?

Results:

Mean age was in years? Please add.

Line 132: There was, not there is (stick to the past tense throughout the whole manuscript, change properly)

To which stiffness values were patients reduced under therapy? You just state the initial values, please add. Thank you.

Line 154: In the study group, => please correct the spelling.

The rate of side effects is statistically significantly lower 156 for locally treated (11.1% vs. 17.6%; p=0.001) and systemically treated patients (33.3% vs. 28.5; p=0.01) in study group

Correct language in this case (see above), thank you. Could you explain, what was the intention to compare and test for significance concerning side effects if it was about the monitoring efficacy of SWE?? It doesn’t seem to make any sense to mention statistics of side effects, which do not support the study results concerning the SWE efficacy, and I’d be careful mentioning these data, as significance can just have been tested in 7 patients, which normally is not enough to calculate such values. I’d recommend to delete this information.

Furthermore, in Table 2, there are just 4 not 5 patients mentioned with side effects in the control arm. How many were there finally?

Check for language overall.

Discussion:

Line 168: Only reduction, without s. Please correct.

Only clinical and sonographic follow-up has been 177 recommended as a treatment option

What does clinical treatment mean? Or clinical follow up? Because clinical and US follow up is not a treatment, this is diagnostics. Please correct properly.

For me as a reader away from this topic in clinical routine, I don’t understand what would be the typical treatment for IGM, antibiotics? Corticosteroids? SSurgery? Is there a guideline? Why did you only look out for coricoid treatment, and not antibiotically treated patients?

Lines 191-201: I’d recommend to shorten this section as focus should be US and not the side effects, treatment or dose management of steroids.

SWE is a relatively new, sonography based, technique used to evaluate tissue stiffness, 203 and its use has been expanding. A region of interest (ROI) can be placed on a colored 204 elastogram to measure tissue stiffness, which can be expressed in meters per second 205 (m/S) or converted to kilopascals (kPa).

These information needed to go in the M&M section

Only two patients (16.6%) were af- 212 fected by our method's failure, and we were forced to revert to the original treatment 213 protocol.

Please explain why did you interpret this result as a failure? What was the failure, that you needed to go back to original dose? I didn’t understand this fact, please explain more in detail and revise this whole paragraph. I also didn’t understand why locally treated patients had an increase in injections?

This is the only similar study found in the literature, and its results were contradic- 229 tory [15]

Why are their results contradictory?

Line 232: the sample size was insufficient to provide a statistically significant result.

But you state in results sections that basically all your results were statistically significant? I’m confused now, please clarify

Line 239 ff: Please improve the English language here, thank you.

Conclusion:

In conclusion, corticosteroid therapy is essential in the treatment of IGM.

This is acutally not a result of your study, delete this.

Figure 1: there are no measurements on the images, is this correct? Because in the corresponding legend measurement values are mentioned, so it would be ideal to adapt the figure in this way as to show the absolute values as well.

Figure 2: The authors should change the font of letters and size of the figure to the font used in the main text.  And systemic dose of 0.8 mg/kg isn’t mentioned in the text, should be added there too.

Very few references, but those are well chosen.

Author Response

Manuscript ID: jcm-2122363

Shear Wave Elastography Correlated Dose Modifying: Can We Reduce Corticosteroid Doses in Idiopathic Granulomatous Mastitis Treatment? Preliminary Results

Dear Reviewer,

We appreciate the time and effort that you dedicated to providing feedback on our manuscript and are grateful for the insightful comments and valuable improvements to our paper. We have incorporated most of the suggestions. Please see below for a point-by-point response to the comments and suggestions. And please see the attachment for the last edited version of the article with track changes enabled format-Word file.

Reviewer #1 COMMENTS:

1) “4%) patients in the study group, corticosteroids doses were reduced and successfully treated. In the study group, the median dose of corticosteroids was found to be significantly lower in patients both locally (p=0.006) and systemic 22 (p=0.007) corticosteroids.”

1a) Hard to understand these sentences, they need improved English language.

1b) First, the doses were reduced and successfully treated, I hope that all patients were successfully treated, but did you mean nevertheless successful? Please add this information.

1c) And I don’t understand the second sentence as well: “In the study group, the median steroid dose was lower compared to control group?? Overall or just during the follow up? Was the dose reduced maybe?? Please clarify.

1a,1b,1c. Response:

In line with your suggestions, we have rewritten the relevant section of the abstract. The revised version is shown below.

“Ten (83.4%) of 12 patients in the study group were successfully treated by reducing corticosteroid doses with follow-up. In 2 (16.6%) of 12 patients, were reverted to the initial treatment protocol, due to an increase in kPa values during the follow-up. Nevertheless, successful treatment results were obtained in these two patients without reducing the corticosteroid dose. When compared to the control group, the median corticosteroid dose in the study group was significantly lower in patients using both local (p <0.01) and systemic (p < 0.01) corticosteroids.”

2) “Side effects due to corticosteroids were found to be statistically significantly lower in the study group”. Statistically significantly is too much, better e.g. found to be significantly lower.

2.Response:

We revised this sentence in line with your suggestions.

“Side effects due to corticosteroids were found to be significantly lower in the study group (p < 0.05).”

3) Introduction: “Long-term high-dose corticosteroid use, on the other hand, has significant adverse 49 effects [3,4]. Considering that, recently, local corticosteroid treatment was demonstrated 50 to be equally effective as systemic corticosteroid therapy [4].”

3a) I’d like to ask the authors, if you could add an information (short only) about how the imaging was performed prior to elastography to monitor either treatment response or the mastitis itself. Thank you.

3a.Response:

To properly diagnose and track the disease, we combine sonographic and elastographic tests. Before beginning elastography, a supine sonographic breast exam was conducted. For future reference, we prefer the radial scanning method, record the lesions' precise location (at breast "o clock") and size. Lesions can be recorded and saved for future reference.

3b) Line 62: Better reduce than diminish

3b. Response:

We revised this sentence in line with your suggestions.

“We have noted that the stiffness values of the healing tissue reduce during the therapy and follow-up phase.”

3c) “the efficacy of SWE stiffness adjusted corticosteroid therapy 65 method in IGM cases.”

Better: the efficacy of SWE for adjustment and monitoring of corticosteroid therapy in IGM cases.

3c. Response:

We revised this sentence in line with your suggestions.

“The current study sought to assess the efficacy of SWE for adjustment and monitoring of corticosteroid therapy in IGM cases”

4) M&Ms:

4a) Could be more detailed overall, there are many information missing, e.g. in the patients section: Patients who received biopsy were included. There are no information about the population (Consecutive patients, only female, male too? How many were excluded due to criteria??

4a. Response:

The material method section was revised and reorganized. In line with your suggestions, we added the gender of the patients, the total number of patients, the number of excluded patients, and the exclusion criteria.

4b) Language needs revision by a native speaker (especially the treatment section, e.g. were injected perilesional areas…that’s not English, injected into would be), same with administered.

4b. Response:  

The relevant paragraph and the whole manuscript were reviewed by a native speaker: “Stable treatment method: Two types of treatment used were used in this study: local and systemic. There were no patients to receive both treatments. Following the drainage of any loculated fluid, a 21-G needle was used to inject 5 mL of saline and 1 mL of a 40 mg methylprednisolone acetate solution into the perilesional regions under sonographic guidance. 0.8 mg/kg/d methylprednisolone acetate was given orally for the systemic treatment. All of the patients were monitored on a monthly basis. The treatment was con-tinued until the patient was completely recovered.”

4c) “The treatment was continued until the patient was completely recovered.” Or until the study was over? How long was it monitored? Or did all included patients recover in this time?

4c. Response:

The following sentence was added into materials and methods section: “All the included patients were followed-up for 2 years and final outcomes were compared.”

4d) Please add the information about the used transducers (MHz?).

4d. Response:

The demanded information was added: “High-frequency linear-array transducers (L12-3, 3-12 MHz) in the longitudinal and transverse planes were used to perform examinations (Affiniti 70g Philips Healthcare, Best, the Netherlands)”

4e) And why did you only measure the kPa and not the Shear wave speed?

4e. Response:

The ultrasound device and the relevant software primarily report the stiffness value as kPa. To create a uniform and understandable design, we have preferred to use only kPa.

4f) How many measurements did you take in each case to assess kPa, did you calculate a mean value of various measurements or just a single measurement?

4f. Response:

We added it in line with your suggestions.

“Elasticity values (kPa) of the lesions were measured three times from the different areas by the same observer, and the average of these measurements was recorded as the final data.”

4g) Where did you measure, same area? Different quadrants of the breast? Just injection side? (Could falsify measurements due to scarring tissue). Please add these information.

4g. Response:

We added following sentences;

“For future reference, we preferred the radial scanning method, recorded the lesions' quadrant, precise location (at breast "o clock"), size and nipple to lesion distance.”

“Due to the tissue changes secondary to the treatment and the time elapsed, it was not possible to match the previous measurement localization exactly. But control scans were performed on the mastitis areas by the same person with the same number of measurements”

5a) Did the control group have IGM or no?

5a. Response:

Yes, all patients in the control group had IGM. We have added this information to the relevant section and revised it as follows.

“Twenty-four patients with a pathological diagnosis of IGM were included in the control group.”

5b) And monthly control scans were performed always with the same number of measurements? Same area?

5b. Response:

Due to the tissue changes secondary to the treatment and the time elapsed, it was not possible to match the previous measurement localization exactly. But yes, control scans were performed on the mastitis areas by the same person with the same number of measurements.

We added the following sentence to the elastographic imaging section.

“Due to the tissue changes secondary to the treatment and the time elapsed, it was not possible to match the previous measurement localization exactly. But control scans were performed on the mastitis areas by the same person with the same number of measurements.”

6) Results:

6a) Mean age was in years? Please add.

6a. Response:

We revised this sentence in line with your suggestions.

“Mean age of the study group was 37.8 ± 2.4 years and control group was 38.1 ± 3.7 years. There was no statistically significant difference between the mean ages.”

6b) Line 132: There was, not there is (stick to the past tense throughout the whole manuscript, change properly)

6b. Response:

We revised this sentence in line with your suggestions.

“Mean age of the study group was 37.8 ± 2.4 years and control group was 38.1 ± 3.7 years, there was no statistically significant difference between the mean ages.”

6c) To which stiffness values were patients reduced under therapy? You just state the initial values, please add. Thank you.

6c. Response:

It is stated in Column 5 of Table 1.

We have revised the title of the relevant column to make it clearer.

“After treatment/before dose adjustment kPa”

6d) Line 154: In the study group, => please correct the spelling.

6d. Response:

We corrected the spelling.

“In study group;” ………….  “In the study group,”

6e) The rate of side effects is statistically significantly lower 156 for locally treated (11.1% vs. 17.6%; p=0.001) and systemically treated patients (33.3% vs. 28.5; p=0.01) in study group

Correct language in this case (see above), thank you.

6e. Response:

We revised this sentence in line with your suggestions.

“The rate of side effects was statistically significantly lower for locally treated (11.1% vs. 17.6%; p < 0.01) and systemically treated patients (33.3% vs. 28.5%; p = 0.01) in the study group (Table 2).”

6f) Could you explain, what was the intention to compare and test for significance concerning side effects if it was about the monitoring efficacy of SWE?? It doesn’t seem to make any sense to mention statistics of side effects, which do not support the study results concerning the SWE efficacy, and I’d be careful mentioning these data, as significance can just have been tested in 7 patients, which normally is not enough to calculate such values. I’d recommend to delete this information.

Furthermore, in Table 2, there are just 4 not 5 patients mentioned with side effects in the control arm. How many were there finally?

6f. Response:

One of the biggest problems with steroid therapy is the side-effect profile. This situation is also emphasized for IGM in previous studies in the literature (e.g Reference 4.    Alper F, Karadeniz E, Güven F, Çankaya BY, Yalcin A, Özden K, et al. Comparison of the Efficacy of Systemic Versus Local Steroid Treatment in Idiopathic Granulomatous Mastitis: A Cohort Study. Journal of Surgical Research. 2022;278:86-92.). For this reason, we predicted that stating the side effect profile of the study population would attract more readers' attention and increase confidence in the results. We felt the need to add these findings, considering that the decrease in the frequency of side effects with the method being introduced also supports the steroid dose-reducing effect of the mentioned method.

The error in Table 2 was developed during the editing phase of the submission phase and has been corrected, thank you.

The low number of patients with side effects was added to the discussion as a limitation based on your recommendation: “The number of patients who develop side effects related to steroid use is relatively limited. For this reason, the effect of the method we have presented on the side effect profile has been evaluated relatively limitedly.”

7) Discussion:

7a) Line 168: Only reduction, without s. Please correct.

7a. Response:

We revised this sentence in line with your suggestions.

“We have revealed that through the defined dose adjustment method significant corticosteroid dose and treatment time reduction could be achieved.”

7b) Only clinical and sonographic follow-up has been 177 recommended as a treatment option

What does clinical treatment mean? Or clinical follow up? Because clinical and US follow up is not a treatment, this is diagnostics. Please correct properly.

7b. Response:

We revised this sentence in line with your suggestions.

“Only clinical and sonographic follow-up has been recommended as a follow-up option [9].”

7c) For me as a reader away from this topic in clinical routine, I don’t understand what would be the typical treatment for IGM, antibiotics? Corticosteroids? SSurgery? Is there a guideline? Why did you only look out for coricoid treatment, and not antibiotically treated patients?

7c. Response:

A standardized treatment method has not yet been presented in the literature. Surgical treatment is becoming less and less preferred method, it is used only for abscess drainage. Antibiotic treatments can be used from time to time in the pre-steroid treatment phase. Steroid therapy is the most prominent and widespread method. The reason why we focus on steroid therapy is that this method is widespread and that only and specifically steroid therapy is applied, not other methods, within the radiology department.

7d) Lines 191-201: I’d recommend to shorten this section as focus should be US and not the side effects, treatment or dose management of steroids.

7d. Response:

We have shortened the relevant section in line with your suggestion.

7e) “SWE is a relatively new, sonography based, technique used to evaluate tissue stiffness, 203 and its use has been expanding. A region of interest (ROI) can be placed on a colored 204 elastogram to measure tissue stiffness, which can be expressed in meters per second 205 (m/S) or converted to kilopascals (kPa).”

These information needed to go in the M&M section

7e. Response:

We have substantially revised and reorganized the material method section and added relevant information.

7f) Only two patients (16.6%) were af- 212 fected by our method's failure, and we were forced to revert to the original treatment 213 protocol.

Please explain why did you interpret this result as a failure? What was the failure, that you needed to go back to original dose? I didn’t understand this fact, please explain more in detail and revise this whole paragraph. I also didn’t understand why locally treated patients had an increase in injections?

7f. Response:

The paragraph was revised as recommended: “In our study, the SWE-correlated active dose modification technique demonstrated prom-ising benefits. Only two patients were forced to revert to the original treatment protocol due to increased kPa values during follow-up. Both of these two patients were receiving local treatment, but this cannot be interpreted that the presented dose modifying method has a higher failure risk for local treatment in comparison with systematic treatment, since the number of systemically treated patients (3 patients) was too small and may not be representative of the actual population. The SWE-correlated dose-modifying technique permitted a substantial reduction in total corticosteroid dose for both systemic and local treatment. Consequently, the incidence of side effects decreased, leading us to hypothesize that the decrease in corticosteroid dose reflects the clinical course. On the other hand, the SWE-correlated dose-modifying technique slightly increased the duration of treatment for locally treated patients. This increase in the time of treatment means that patients will get more injections. As a result, despite a decrease in total dose and correlated side effects, pa-tient compatibility could be negatively impacted by this dose modifying.

7g) This is the only similar study found in the literature, and its results were contradic- 229 tory [15]

Why are their results contradictory?

7g. Response:

The relevant paragraph was revised as recommended. This sentence was added: “We believe that the main reason for the contradicting results is that patients in Aslan et al.'s study were treated with a variety of ways, not just with steroids. This can change the outcome of the treatment and produce results that contradict ours.”

7h) Line 232: the sample size was insufficient to provide a statistically significant result. But you state in results sections that basically all your results were statistically significant? I’m confused now, please clarify

7h. Response:

The sentence was rewritten in a more understandable manner, as recommended: “In two patients, the presented method was ineffective. Even though these patients were older than the mean age of the population, two patients are insufficient to draw a judgment concerning the unfavorable effect of age on the treatment approach.”

7i) Line 239 ff: Please improve the English language here, thank you.

7i. Response:

The English language of the mentioned paragraph was revised: “There are some limitations worth noting about the current study. The most significant limitation is the small sample size, especially for the systemic treatment group. The sample size is only sufficient for a pilot study. Additional studies with larger populations and more systemically treated subjects would reveal different results. Additional studies with larger populations and more systemically treated subjects are needed. As a further consequence of the small sample size, all of the included patients were successfully treated, and no recurrent cases were observed. We were unaware of the effect of the presented method on cases of this nature. Due to the small number of patients for whom our method failed, we are unable to identify any potential flaws in our approach. Our clinical experiences were comprised of the treatment doses and rate of decrease in kPa values to adjust the new dosing regimen. Results may vary depending on different dosage and protocol. Due to the fact that each SWE examination was conducted by the same researcher, we cannot provide data on inter-rater variability. The number of patients who develop side effects related to steroid use is relatively small. For this reason, the effect of the method we have presented on the side effect profile has been evaluated relatively limited.”

8) Conclusion:

8) In conclusion, corticosteroid therapy is essential in the treatment of IGM.

This is acutally not a result of your study, delete this.

8.Response:

We deleted this sentence and revised conclusion section in line with your suggestions.

“In conclusion the presented SWE-correlated dose-modifying technique may reduce corticosteroid doses and corticosteroid related side effects without compromising treatment efficacy significantly.”

9) Figure 1: there are no measurements on the images, is this correct?

Because in the corresponding legend measurement values are mentioned, so it would be ideal to adapt the figure in this way as to show the absolute values as well.

9.Response:

Since we presented by combining B mode images and elastographic images in figure 1, and we gave subtitles to each image for this reason, the cutting process was performed on the images. For this reason, the relevant data are detailed in the figure legend.

10) Figure 2:

10a) The authors should change the font of letters and size of the figure to the font used in the main text. 

10a. Response:

We changed the font and size of the figure. The revised version is shown below.

10b) And systemic dose of 0.8 mg/kg isn’t mentioned in the text, should be added there too.

10b. Response:

We wrote this sentence in the M&M/treatment section.

“For the systemic treatment, 0.8 mg/kg/d methylprednisolone acetate was given orally.”

11) Very few references, but those are well chosen.

11.Response:

Thank you very much for all your valuable comments, suggestions, contributions and efforts.

Reviewer 2 Report

*Introduction*

Line 55-56: “SWE is slower, more difficult to learn, more expensive, and less common than strain elastography” - This is not a relevant comparison and should be excluded from the article.

Line 62, 63, 89, 124, 206, 224, 225 : “stiffness values” -> terminology; throughout the text several different terms are used and it is confusing-> stiffness value, stiffness, SWE stiffness, elasticity, kPa value, rigidity… ; decide for one term only -> strain modulus (Look at: Snoj et al. 2020, Ultrasound Elastography in Musculoskeletal Radiology: Past, Present, and Future)

Line 62: “diminish” use a different word as diminish is too “strong” word in this case

*Methods*

*I would suggest considering changing some subtitles in the Methods and rearranging them more systematically. *

Systematic description of control and study groups should be added. It must be clear how many groups were there, how participants were selected to groups, randomization of patients, what was the intervention in each group,… In this article the group details are not clearly stated. Lack of details regarding the groups in the Methods section makes it more difficult to follow the results.

Procedure that you used for elastography measurements should also be clearly described (detailed description of ROI location and size,…). Information in Methods should be detailed enough to allow the reader to replicate the same research again based only on the info in the article. In this article this is not the case. With more systematically structured methods most of the mentioned issues could be solved.

Line 71: no need to use adjective “pathological” diagnosis. Number and mean and SD of age and other necessary information about participants is missing in this section

Line 73: “Exclusion criteria included…” It is usually stated what it is excluded not “included in the exclusion…”

Line 75,76: “(12 patients)” -> number of patients that dropped out of the study should be reported in the results section. In methods it should only be mentioned the number of participants and the mean and SD information for the anthropometric or other information important for the study

Line 77: “two types of treatment used” -> it should be explained how the participants were divided in the intervention and control group, how was determined who received local and who systemic treatment. Randomization of the patients in 2 groups should be explained

Line 87: rephrase and add details how the ROI was determined

Line 88-89: “care was taken to avoid applying pressure to the probe 88 and to keep the practitioner's hand fixed” -> this is how the method is used in clinic or in practice but for the research purpose the repeatability and consistency of measurements is low as it depends on subjective perception of the “pressure applied” to the probe and of the “fixed hand”

Line 89: “Elasticity is measured in kPa” -> it is not completely true as shear modulus is measured in kPa -> look into Snoj et al. 2020 and other literature for detailed definition of shear modulus, Young modulus, and correct use of terminology

Line 89,129 136: “Elasticity is….” The article reports the research that happened in the past so the text should be in the past tense. Revise the article for grammar and correct use of tense

Line 90-91: Usually the tissue is measured three times on the same location by the same observer and the mean of the measurements is used. So in your case you should do three measurements on each of the three locations to improve the accuracy of the measurements

Line 93-94: Locations of the measurements should be described more accurately in terms of pre and post intervention. Location of the pre intervention measurement should be marked so that the post 1 month and end of research measurement could be performed on exactly the same location as the pre-testing measurement. How did you make sure that the pre and post readings were on exactly the same location? In your research report it is only stated the same granulomatous mastitis was measured pre and post but the size and the shape can change with successful treatment in months so the exact location and perspective of the tissue cannot be replicated without precisely determined location of the measurement.

Line 94-95: “A reduction of 20% or more in kPa values was considered a positive response to treatment” -> how was this determined? Was this threshold 20% supported in the literature? Also, if you determine the threshold 20% then the values above the threshold are not only “considered” but they are “defined”(or synonym that is stronger than considered -> determined) as positive response

Line 99: literature supporting 0.6mg/kg

Line 103: Literature supporting 0.4mg/kg

Line 106-110: Information about control and intervention group should be added to the Participants section

Line 11-120: p-value was set at p<0.01, p<0.05,…? And confidence interval 95%,99%

Line 122-123: more detailed description of the lesion location and size could be provided. If the lesion size decreased during the research, how can you be certain that the post treatment measurement was performed on the same location?

Line 125: add kPa next to the numbers

Figure1: The image should also include the information regarding the range of the colour scale on the right side (For example: blue is 0kPa, red is 90kPa). Other details of the elastography set-up should be mentioned here and/or in the methods.

Figure 2: Between Level 1 and Level 2 -> no progression or cessation in the Level 4 Right side

- Level 6 and Level 7 -> for the purpose of transparency of the scheme and to make it more logical I would suggest changing the boxes on the right (“Positive treatment response” and “Continue with the same protocol”) and put them on the left side of the “Progression or cessation of treatment response” and “Return to the initial treatment protocol”

*Results*

Line 129-131: No need to use “current” it is clear that you talk about the results of this “current” study and not some other study. In the Methods section line 75-76 you mentioned that 12 patients dropped out during the research for various reasons and it is should be mentioned here in the results

Line 132: Number of participants and mean and SD of age should be reported in the methods not only in the results[JP1] <#_msocom_1>

Line 143-158: The number od decimal spaces used should be the same throughout the article. Usually, 2 decimal spaces is enough. You can also report p-value as p<0.05 or p<0.01 for example.[JP2] <#_msocom_2>

Line 154: “side effects” -> determine side effects in methods. Side effects could be added in subheading treatment Line 77-83

Line 156: Please double check the significance of the difference between the number of participants experiencing the side effects in control vs study group.

Line 158: “28.5%”

Table 2: How was side effect “weight gain” determined strictly as a side effect of your intervention and the patient did not gain weight due to other reasons. Why were other possible reasons excluded. Especially, since the study duration was 2 years and in some side effects mentioned, can be due to several other reasons.

*Discussion*

*There are some sectons that should not be found in the discussion but maybe in the introduction*

Line 166: “practicability of corticosteroid dose” -> were other factors considered or how they were excluded?

Line 170-175: This paragraph is more suitable for the Introduction section -> information in the Lines 170-171 is already mentioned in the Introduction

Line 176-186: This section is not suitable for the Discussion as there is no connection with the study results

Line 203-210: Not suitable for the Discussion section

Line 211: “According to our first and preliminary findings” -> unnecessary, “In our study” should be enough

Line 212: “16.6%” No need to report numbers in discussion section. Only the summary f the numbers mentioned in the Results section is enough

Line 213: “method’s failure” ???

*Conclusion*

Line 250: IT is not a direct summary of the study findings. A strict direct answer to your study hypothesis should be added her and no additional info that were not directly investigated in this study should be added.

Line 251-252: “According to our first 251 and preliminary results” -> not needed.

Author Response

Manuscript ID: jcm-2122363

Shear Wave Elastography Correlated Dose Modifying: Can We Reduce Corticosteroid Doses in Idiopathic Granulomatous Mastitis Treatment? Preliminary Results

Dear Reviewer,

We appreciate the time and effort that you dedicated to providing feedback on our manuscript and are grateful for the insightful comments and valuable improvements to our paper. We have incorporated most of the suggestions. Please see below for a point-by-point response to the comments and suggestions. And please see the attachment for the last edited version of the article with track changes enabled format-Word file.

1) *Introduction*

Line 55-56: “SWE is slower, more difficult to learn, more expensive, and less common than strain elastography” - This is not a relevant comparison and should be excluded from the article.

1.Response:

In line with your suggestion, we removed this sentence from the article.

2) Line 62, 63, 89, 124, 206, 224, 225 : “stiffness values” -> terminology; throughout the text several different terms are used and it is confusing-> stiffness value, stiffness, SWE stiffness, elasticity, kPa value, rigidity… ; decide for one term only -> strain modulus (Look at: Snoj et al. 2020, Ultrasound Elastography in Musculoskeletal Radiology: Past, Present, and Future)

2.Response:

We reviewed the article you suggested and corrected the terminology as much as possible. We used the terms “elasticity value (kPa)” and “stiffness” more than before.

3) Line 62: “diminish” use a different word as diminish is too “strong” word in this case

3.Response

In line with your suggestion, we used the word “reduce” instead of “diminish”.

4) *Methods*

*I would suggest considering changing some subtitles in the Methods and rearranging them more systematically. *

4.Response

We have rearranged and revised the methods section in line with your suggestions.

The new subtitle format is as follows.

Materials and Methods… Participants… Patients group:… Control group:… The exclusion criteria…. Elastographic imaging technique:… Treatment… Stable treatment method:… Assessment of treatment response and study protocol (dose-modifying technique):… Side effects:… Statistical analysis:

5) Systematic description of control and study groups should be added. It must be clear how many groups were there, how participants were selected to groups, randomization of patients, what was the intervention in each group,… In this article the group details are not clearly stated. Lack of details regarding the groups in the Methods section makes it more difficult to follow the results.

5.Response

We have rearranged and revised the materials and methods section in line with your recommendations 4 and 5.

6) Procedure that you used for elastography measurements should also be clearly described (detailed description of ROI location and size,…). Information in Methods should be detailed enough to allow the reader to replicate the same research again based only on the info in the article. In this article this is not the case. With more systematically structured methods most of the mentioned issues could be solved.

6.Response

We reviewed the elastographic measurements section again and made some corrections and additions.

7) Line 71: no need to use adjective “pathological” diagnosis. Number and mean and SD of age and other necessary information about participants is missing in this section

7.Response

We deleted the word "pathological" in line with your suggestion and added information about participants to this section.

“Patients who received a pathological diagnosis of IGM from a tru-cut biopsy and were willing to participate were included in the study.”

“A total of 36 female participants were included in the study. The mean age of the participants was 38.0 ± 3.4 years.”

8) Line 73: “Exclusion criteria included…” It is usually stated what it is excluded not “included in the exclusion…”

8.Response

We have reviewed the exclusion criteria and rearranged them in line with your suggestion.

“The exclusion criteria of this study were any contraindication for systemic and/or local corticosteroid use, malignancy, immunosuppression, being in the active lactation period, cessation of treatment due to the patient's will or for medical reasons, and lack of follow-up data (12 patients).”

9) Line 75,76: “(12 patients)” -> number of patients that dropped out of the study should be reported in the results section. In methods it should only be mentioned the number of participants and the mean and SD information for the anthropometric or other information important for the study

9.Response

The words “12 patients” were removed from this section and added to the Results section with the following sentence.

“According to the exclusion criteria of the study, 12 patients were excluded.”

10) Line 77: “two types of treatment used” -> it should be explained how the participants were divided in the intervention and control group, how was determined who received local and who systemic treatment. Randomization of the patients in 2 groups should be explained

10.Response

We have rearranged and revised the materials and methods section in line with the recommendations.

11) Line 87: rephrase and add details how the ROI was determined

11.Response

We have rearranged and revised the materials and methods section in line with the recommendations.

12) Line 88-89: “care was taken to avoid applying pressure to the probe 88 and to keep the practitioner's hand fixed” -> this is how the method is used in clinic or in practice but for the research purpose the repeatability and consistency of measurements is low as it depends on subjective perception of the “pressure applied” to the probe and of the “fixed hand”

12.Response

Applied pressure values were added according to the suggestions.

13) Line 89: “Elasticity is measured in kPa” -> it is not completely true as shear modulus is measured in kPa -> look into Snoj et al. 2020 and other literature for detailed definition of shear modulus, Young modulus, and correct use of terminology

13.Response

The relevant literature was examined and the suggested changes were made:

“We expressed the elasticity data in kilopascals (kPa), which is derived SI (system international) unit of shear modulus. Shear modulus can be defined as the ratio of shear stress and shear strain.”

14) Line 89,129 136: “Elasticity is….” The article reports the research that happened in the past so the text should be in the past tense. Revise the article for grammar and correct use of tense

14.Response

In line with your suggestion, we reviewed the article for tense errors and made some corrections.

Elasticity is measured …… Elasticity was measured

The current study includes 36 female patients…… The current study included 36 female patients

There are 12 patients…. There were 12 patients

15) Line 90-91: Usually the tissue is measured three times on the same location by the same observer and the mean of the measurements is used. So in your case you should do three measurements on each of the three locations to improve the accuracy of the measurements

15.Response

In line with your suggestions, materials and methods section was revised in a more clear manner.

16) Line 93-94: Locations of the measurements should be described more accurately in terms of pre and post intervention. Location of the pre intervention measurement should be marked so that the post 1 month and end of research measurement could be performed on exactly the same location as the pre-testing measurement. How did you make sure that the pre and post readings were on exactly the same location? In your research report it is only stated the same granulomatous mastitis was measured pre and post but the size and the shape can change with successful treatment in months so the exact location and perspective of the tissue cannot be replicated without precisely determined location of the measurement.

16.Response

Due to the tissue changes secondary to the treatment and the time elapsed, it was not possible to match the previous measurement localization exactly. But, control scans were performed on the mastitis areas by the same person with the same number of measurements.

We added the following sentence to the elastographic imaging section.

“Due to the tissue changes secondary to the treatment and the time elapsed, it was not possible to match the previous measurement localization exactly. But control scans were performed on the mastitis areas by the same person with the same number of measurements.”

17) Line 94-95: “A reduction of 20% or more in kPa values was considered a positive response to treatment” -> how was this determined? Was this threshold 20% supported in the literature? Also, if you determine the threshold 20% then the values above the threshold are not only “considered” but they are “defined”(or synonym that is stronger than considered -> determined) as positive response

17.Response

The 20% reduction criterion is based on our clinical experience and observations, not a literature data. The mentioned sentence was also added to the materials and methods section.

“(The 20% reduction criterion is based on our clinical experience and observations, not a literature data)”

In line with your suggestion, we deleted the word "considered" and replaced it with the word "determined".

18) Line 99: literature supporting 0.6mg/kg

19) Line 103: Literature supporting 0.4mg/kg

18-19.Response

We added the reference to the relevant lines.

20) Line 106-110: Information about control and intervention group should be added to the Participants section

20.Response

The materials and methods section was completely revised according to the comments to be more clear.

21) Line 11-120: p-value was set at p<0.01, p<0.05,…? And confidence interval 95%,99%

21.Response

We added the following sentence to the end of the statistical analysis section.

“A value of P < 0.05 was regarded as statistically significant.”

22) Line 122-123: more detailed description of the lesion location and size could be provided. If the lesion size decreased during the research, how can you be certain that the post treatment measurement was performed on the same location?

22.Response

Thank you for emphasizing this important point.

Due to the tissue changes secondary to the treatment and the time elapsed, it was not possible to match the previous measurement localization exactly. But, control scans were performed on the mastitis areas by the same person with the same number of measurements.

We added the following sentence to the elastographic imaging section.

“Due to the tissue changes secondary to the treatment and the time elapsed, it was not possible to match the previous measurement localization exactly. But control scans were performed on the mastitis areas by the same person with the same number of measurements.”

23) Line 125: add kPa next to the numbers

23.Response

We added “kPa” next to the numbers.

24) Figure1: The image should also include the information regarding the range of the colour scale on the right side (For example: blue is 0kPa, red is 90kPa). Other details of the elastography set-up should be mentioned here and/or in the methods.

24.Response

More details were added to the materials and methods section about elastographic examination.

Unfortunately, patients’ information overlapped with the range data and as a result, we had to cut all of the information in order to create an anonymous image. Also, subtitles such as a, b, c, and d encased the range of data, too.

25) Figure 2: Between Level 1 and Level 2 -> no progression or cessation in the Level 4 Right side

26) - Level 6 and Level 7 -> for the purpose of transparency of the scheme and to make it more logical I would suggest changing the boxes on the right (“Positive treatment response” and “Continue with the same protocol”) and put them on the left side of the “Progression or cessation of treatment response” and “Return to the initial treatment protocol”

25-26.Response

We changed the design of figure 2 in line with your suggestions. We have added the new version below.

27) *Results*

Line 129-131: No need to use “current” it is clear that you talk about the results of this “current” study and not some other study. In the Methods section line 75-76 you mentioned that 12 patients dropped out during the research for various reasons and it is should be mentioned here in the results

27.Response

We deleted the word “Current”. We mentioned “12 excluded patients” in this section.

“The current study”….. “The study”

“According to the exclusion criteria of the study, 12 patients were excluded.”

28) Line 132: Number of participants and mean and SD of age should be reported in the methods not only in the results[JP1] <#_msocom_1>

28.Response

In line with your suggestions, we added "Number of participants and mean and SD of age" to the relevant place in the methods section.

29) Line 143-158: The number od decimal spaces used should be the same throughout the article. Usually, 2 decimal spaces is enough. You can also report p-value as p<0.05 or p<0.01 for example.[JP2] <#_msocom_2>

29.Response

We reviewed the number of decimal spaces used and rearranged them to be 2 decimals.

30) Line 154: “side effects” -> determine side effects in methods. Side effects could be added in subheading treatment Line 77-83

30.Response

“Side effects: Steroid use-related side effects (Nausea, joint pain, weight gain, edema) during treatment have also been noted.”

The sentence was added to the suggested part.

31) Line 156: Please double check the significance of the difference between the number of participants experiencing the side effects in control vs study group.

31.Response

There was a mistake about the percentage and the number of patients revealing side effects; the mistake in table 2 was corrected.

32) Line 158: “28.5%”

32.Response

We added it.

33) Table 2: How was side effect “weight gain” determined strictly as a side effect of your intervention and the patient did not gain weight due to other reasons. Why were other possible reasons excluded. Especially, since the study duration was 2 years and in some side effects mentioned, can be due to several other reasons.

33.Response

We did not independently decided the side effects. If the relevant clinician (a general surgeon) agreed with us that the newly developed condition was a steroid-related side effect, we accepted this effect as a side effect. The decision for the presence of side effects was made after long-term clinician observation.

34) *Discussion*

*There are some sectons that should not be found in the discussion but maybe in the introduction*

35) Line 166: “practicability of corticosteroid dose” -> were other factors considered or how they were excluded?

36) Line 170-175: This paragraph is more suitable for the Introduction section -> information in the Lines 170-171 is already mentioned in the Introduction

37) Line 176-186: This section is not suitable for the Discussion as there is no connection with the study results

38) Line 203-210: Not suitable for the Discussion section

34-38.Response

In line with your suggestions, we revised the relevant paragraphs in discussion section and made new arrangements discussion and introduction sections. And we modified the relevant sentence.

39) Line 211: “According to our first and preliminary findings” -> unnecessary, “In our study” should be enough

39.Response

We changed it as “In Our Study” in line with your suggestions.

40) Line 212: “16.6%” No need to report numbers in discussion section. Only the summary f the numbers mentioned in the Results section is enough

40.Response

We deleted “16.6%” in line 212.

41) Line 213: “method’s failure” ???

41.Response

The relevant paragraph was revised: “In our study, the SWE-correlated active dose modification technique demonstrated promising benefits. Only two patients were forced to revert to the original treatment protocol due to increased kPa values during follow-up. Both of these two patients were receiving local treatment, but this cannot be interpreted that the presented dose modifying method has more failure risk for local treatment in comparison with systematic treatment, since the number of systemically treated patients (3 patients) was too small and may not be representative of the actual population. The SWE-correlated dose-modifying technique permitted a substantial reduction in total corticosteroid dose for both systemic and local treatment. Consequently, the incidence of side effects decreased, leading us to hypothesize that the decrease in corticosteroid dose reflects the clinical course. On the other hand, the SWE-correlated dose-modifying technique slightly increased the duration of treatment for locally treated patients. This increase in the time of treatment means that patients will get more injections. As a result, despite a decrease in total dose and correlated side effects, patient compatibility could be negatively impacted by this dose modifying.

42) *Conclusion*

Line 250: IT is not a direct summary of the study findings. A strict direct answer to your study hypothesis should be added her and no additional info that were not directly investigated in this study should be added.

43)Line 251-252: “According to our first 251 and preliminary results” -> not needed.

42-43.Response

We have revised the conclusion part in line with your suggestions.

"In conclusion the presented SWE-correlated dose-modifying technique may reduce corticosteroid doses and corticosteroid related side effects without compromising treatment efficacy significantly."

Thank you very much for all your valuable comments, suggestions, contributions and efforts.

Reviewer 3 Report

This contribution presents a valuable both clinically and scientifically relevant proposal of elatrography stiffness to optimize the treatment of mastitis. However, prior to publication, a number of issues should be considered:

- In methods, since one of the main limitations hindering the widespread of SWE for clinical diagnosis is a clear protocol, in the present methods section, a much more detailed protocol of measurement should be described, includind the size of the ROI, the number of ROIs averaged, the choice of the plane orientation of imaging and the exact position, for reproducibility, as well as the applied pressure with the probe, which is known to alter the results.

- Could the healthy breast be used as a control?

- Some references directly related to the objective of the article are missing, and importantly, some of them present matching conclusions with this manuscript, which should be commented in the introduction or discussion: 

Why Are Viscosity and Nonlinearity Bound to Make an Impact in Clinical Elastographic Diagnosis?

G Rus, I H Faris, J Torres, A Callejas, J Melchor

Sensors 20 (8), 2379 (2020)

In view of the quality of the work and the document, it is recommended to be considered for publication conditioned to correction of the points above.

Author Response

Manuscript ID: jcm-2122363

Shear Wave Elastography Correlated Dose Modifying: Can We Reduce Corticosteroid Doses in Idiopathic Granulomatous Mastitis Treatment? Preliminary Results

Dear Reviewer,

We appreciate the time and effort that you dedicated to providing feedback on our manuscript and are grateful for the insightful comments and valuable improvements to our paper. We have incorporated most of the suggestions. Please see below for a point-by-point response to the comments and suggestions. And please see the attachment for the last edited version of the article with track changes enabled format-Word file.

1) This contribution presents a valuable both clinically and scientifically relevant proposal of elatrography stiffness to optimize the treatment of mastitis. However, prior to publication, a number of issues should be considered:

- In methods, since one of the main limitations hindering the widespread of SWE for clinical diagnosis is a clear protocol, in the present methods section, a much more detailed protocol of measurement should be described, includind the size of the ROI, the number of ROIs averaged, the choice of the plane orientation of imaging and the exact position, for reproducibility, as well as the applied pressure with the probe, which is known to alter the results.

1.Response:

In line with your suggestions, we added the following sentences to the method section and revised.

“Measurements were made with 3 different ROIs of 1 cm2, and the average was taken.”

“Examinations were performed with the patient in the supine position.”

“Elastographic examinations were performed on axial images.”

“Elastographic imaging technique: High-frequency linear-array transducers (L12-3, 3-12 MHz) in the longitudinal and transverse planes were used to perform examinations (Affiniti 70g Philips Healthcare, Best, the Netherlands) (Figure 1). The examinations were carried out in a supine position. Elastography values were determined automatically by the device's SWE feature. During ultrasonographic imaging, care was taken to avoid applying pressure to the probe and to keep the practitioner's hand fixed. Elastography values were obtained with a maximum shear wave velocity cut-off of 5 m/s (80 kPa) to improve specificity. We expressed the elasticity data in kilopascals (kPa), which is derived SI unit of shear modulus. Shear modulus can be defined as the ratio of shear stress and shear strain. Elasticity value can be expressed in meters per second (m/S) or converted to kilopascals (kPa). Elasticity values (kPa) of the lesions were measured three times with three different ROIs of 1 cm2 from the different areas by the same observer, and the average of these measurements was recorded as the final data. For future reference, we preferred the radial scanning method, and recorded the lesions' quadrant, precise location (at breast "o clock"), size and nipple to lesion distance. Due to the tissue changes secondary to the treatment and the time elapsed, it was not possible to match the previous measurement localization exactly. But control scans were performed on the mastitis areas by the same person with the same number of measurements.”

2) - Could the healthy breast be used as a control?

2.Response:

We did not include healthy individuals in the study, as our main aim was to test the elastography-mediated dose modulation technique. We used literature data to support the usability of elastography in the breast. For these reasons, we believe that including healthy individuals will not change the results.

3) - Some references directly related to the objective of the article are missing, and importantly, some of them present matching conclusions with this manuscript, which should be commented in the introduction or discussion:

Why Are Viscosity and Nonlinearity Bound to Make an Impact in Clinical Elastographic Diagnosis?

G Rus, I H Faris, J Torres, A Callejas, J Melchor

Sensors 20 (8), 2379 (2020)

3.Response:

Thank you for your suggestion and warning. We used the mentioned reference.

4) In view of the quality of the work and the document, it is recommended to be considered for publication conditioned to correction of the points above.

4.Response:

Thank you very much for all your valuable comments, suggestions, contributions and efforts.

Round 2

Reviewer 1 Report

Dear authors,

thank you for submitting your revised manuscript. I think you've done a great job with your revisions and the revisions improved your manuscript immensely. I congratulate you on your interesting study. 

Kind regards.

Author Response

Thank you for your encouraging comments and efforts.

Reviewer 2 Report

English must be improved.

Please better define the patient selection. It is still very confusing to read. If I understand correctly you had 48 pts and based on exclusion criteria 36 pts remained. How were inital 48 pts gathered?

I still do not undestand the division of groups. There are 4 groups. What is the difference between the study and control group and how were the patients divided? Maybe make another flowchat on pts selection and grouping.

Each patient had 24 follow-ups? Is that correct? I would find it extremely hard to believe that compliance was 100%.

There has been some improvement in describin elastographic methods but it is still unclear. Do not why it is important to mention m/s if it was not used in the study. Please make it straightforward to the readers. Please refer to Urban MW, Nenadic IZ, Chen S, Greenleaf JF. Discrepancies in reporting tissue material properties. J Ultrasound Med 2013. Still terminology issues – somewhere you use kPa values, somewhere elasticity values, somewhere you you both - unify

Only one image was obtained and on this image 3 measurements wre made? Or on 3 seperarte images?

Do not undestand what would for future reference mean. The methods are intended to describe them in detail in order another researcher would like to reaaply the same methodology. Please describe how you have done it.

For injection – was it a single depost or multiple deposit?

Completely recovered should be defined.

Why did you not select 10% or 30% cut off. This is a very vague description. Perhaps you can find some references in literature from treating tendinopathies?

No need to repeat pts number in results. Do not repeat throughout text – in results you again mention 12 pts excluded and repeat avg age etc...

In methods you describe if pregression/cessation. In results you mention recurrence – what is the differenec? Please define.

Amongst 12 patients, 2 patients (Table 1, patients 8 and 9) experienced 284 increased kPa values during follow-up and we returned to the initial treatment protocol 285 (2/12, 16.6%), these patients were all receiving local treatment. – repetition of results in same sentance

When was the treatment finished? Please define this as the follow up is different than weeks of treatment

In methods section please describe how the clinical data has been obtained – the date on side effects.

Paragraphs 2-5 do not fit in discussion. These would fit better with introduction. Discussion is a section of the paper where your results are discussed with the results from the literature- as you have in paragraphs 6-8

When interpreting side effects – would you not consider it unfair to put all side effects for local and sistemic in single group. I am afraid I see a great bias here that can be easily misinterpreted. This is a major flaw. Please comment on that and restructure the text accordingly.

Author Response

Manuscript ID: jcm-2122363

Shear Wave Elastography Correlated Dose Modifying: Can We Reduce Corticosteroid Doses in Idiopathic Granulomatous Mastitis Treatment? Preliminary Results

Dear Reviewer,

We appreciate the time and effort that you dedicated to providing feedback on our manuscript and are grateful for the insightful comments on and valuable improvements to our paper. We have incorporated most of the suggestions. Please see below for a point-by-point response to the comments and suggestions.

COMMENTS:

1) Please better define the patient selection. It is still very confusing to read. If I understand correctly you had 48 pts and based on exclusion criteria 36 pts remained. How were inital 48 pts gathered?

1.Response:

The patients subheading was reorganized and revised according to the comments:

“During the study period mentioned above, 48 patients with biopsy-confirmed diagnosis of granulomatous mastitis, who could be treated with steroids, and who agreed to participate in the study were included. The exclusion criteria of this study were any contraindication for systemic and/or local corticosteroid use, malignancy, immunosuppression, being in the active lactation period, cessation of treatment due to the patient's will or for medical reasons, and lack of follow-up data. According to the exclusion criteria of the study, 12 patients were excluded.

A total of 36 female patients who received a diagnosis of IGM from a tru-cut biopsy were included in the study. The mean age of the patients was 38.0 ± 3.4 years.”

2) I still do not undestand the division of groups. There are 4 groups. What is the difference between the study and control group and how were the patients divided? Maybe make another flowchat on pts selection and grouping.

2.Response:

We have created a flowchart according to the recommendation (Figure 1)

Figure 1. Grouping of participants and treatment methods

3) Each patient had 24 follow-ups? Is that correct? I would find it extremely hard to believe that compliance was 100%.

3.Response:

The compliance of the patients was not 100%. The noncompliant ones were excluded. Even though the rate of compliance can seem to be high, the main reason is that the patients were mainly diagnosed and treated in our clinic, also most of them came for routine breast screening, too. In other words, most of the patients were familiar with our breast clinic, and this situation increased the compliance rates.

4) There has been some improvement in describin elastographic methods but it is still unclear. Do not why it is important to mention m/s if it was not used in the study. Please make it straightforward to the readers. Please refer to Urban MW, Nenadic IZ, Chen S, Greenleaf JF. Discrepancies in reporting tissue material properties. J Ultrasound Med 2013. Still terminology issues – somewhere you use kPa values, somewhere elasticity values, somewhere you you both - unify

4.Response:

We added this sentence (containing m/s) with the suggestion of the other reviewer. In line with your suggestions, we have revised the article in terms of this terminology. We made changes in the form of “elasticity value” in approximately 20-25 different places. 

5) Only one image was obtained and on this image 3 measurements wre made? Or on 3 seperarte images?

5.Response:

As we tried to express as “from the different areas”, measurements were made from different affected areas, hence from different images.

6) Do not undestand what would for future reference mean. The methods are intended to describe them in detail in order another researcher would like to reaaply the same methodology. Please describe how you have done it.

6.Response:

We have revised the mentioned section according to the comment: “We used the radial scanning method both to find the lesion easily in later examinations and for a different practitioner to understand the described localization clearly and accurately. We recorded the lesions' quadrant, precise location (at breast "o clock"), size and nipple to lesion distance.”

7) For injection – was it a single depost or multiple deposit?

7.Response:

After the methylprednisolone solution was prepared, it was injected into the perilesional areas from different localizations.

Line 111-114: “Following the drainage of any loculated fluid, a 21-G needle was used to inject 5 mL of saline and 1 mL of a 40 mg methylprednisolone acetate solution into the perilesional regions under sonographic guidance.”

8) Completely recovered should be defined.

8.Response:

“(complete recovery: no recurrence during 2 year period of follow-up).” This sentence was added into the relevant section.

9) Why did you not select 10% or 30% cut off. This is a very vague description. Perhaps you can find some references in literature from treating tendinopathies?

9.Response:

We have added the following explanation as recommended:

“Since this one is the first study in the literature to examine the dose-modifying method via SWE, we cannot find a similar and/or exemplary study in the literature to define a more objective reduction percentage)”

10) No need to repeat pts number in results. Do not repeat throughout text – in results you again mention 12 pts excluded and repeat avg age etc...

10.Response:

We agree with you; however they were added after a recommendation from another reviewer. As a result, We respectfully request that the specified parts remain as they are.

11) In methods you describe if pregression/cessation. In results you mention recurrence – what is the differenec? Please define.

11.Response:

Line 131-132: “If progression or cessation of treatment response (increased or similar control kPa values) occurred in any phase of follow-up, the initial dosing regimen was reinstated [4] (Figure 2).”

As the definition “progression or cessation of treatment response”, we meant a failure of the dose modifying technique during the treatment phase. By saying recurrence, we meant the recurrence of the pathology during 2 year follow-up period after a complete recovery. We have added the explanation into the results section.

12) When was the treatment finished? Please define this as the follow up is different than weeks of treatment

12.Response:

The treatment was finished when the lesions were vanished.  The treatment occurred in different times in different patients. Yet, we have continued the follow-up for 2 years for all patients independent from the duration of treatment, in order to detect any recurrence. As a result, follow-up period is different from the duration of treatment. We have written the sentence into the materials and methods section: “All the included patients were followed up for 2 years and final outcomes were compared.”

13) In methods section please describe how the clinical data has been obtained – the date on side effects.

13.Response:

The following sentence was added according to the comments: “The patients were evaluated for the presence of steroid-related side effects during imaging controls with the participation of the relevant clinician.”

14) Paragraphs 2-5 do not fit in discussion. These would fit better with introduction. Discussion is a section of the paper where your results are discussed with the results from the literature- as you have in paragraphs 6-8

14.Response:

Even though we agree with the reviewer, most of the content of the mentioned paragraphs were added as a result of the suggestion of another reviewer towards summarizing general data about the treatment of idiopathic granulomatous mastitis, and the negative effects of steroid use. So, we respectfully request that the specified parts remain as they are.

15) When interpreting side effects – would you not consider it unfair to put all side effects for local and sistemic in single group. I am afraid I see a great bias here that can be easily misinterpreted. This is a major flaw. Please comment on that and restructure the text accordingly.

15.Response:

In fact, firstly, we compared the study and control group without looking at the treatment type (local or systemic steroid). Then, we compared the side effects in local and systemic treatment separately between the study and control groups by adding the treatment type (local or systemic steroid). But, we accept that the number of side effects and our evaluation are not sufficient due to the small number of patients. However, since the primary purpose of this dose-modifying treatment protocol was to reduce corticosteroid doses, we could not ignore the side effects. We hope that the limitation in this matter will be tolerated by the esteemed referee. We have revised the relevant sentence according to the recommendation: “Consequently, even though the total number of patients experiencing side effects is quite limited, the incidence of side effects decreased, leading us to hypothesize that the decrease in corticosteroid dose reflects the clinical course.”

Thank you for your encouraging comments and efforts.
